# Molecular Understanding of the Interfacial Interaction and Corrosion Resistance between Epoxy Adhesive and Metallic Oxides on Galvanized Steel

**DOI:** 10.3390/ma16083061

**Published:** 2023-04-13

**Authors:** Shuangshuang Li, Yanliang Zhao, Hailang Wan, Jianping Lin, Junying Min

**Affiliations:** 1School of Mechanical Engineering, Tongji University, Shanghai 201804, China; 2Baoshan Iron & Steel Co., Ltd., Shanghai 201900, China

**Keywords:** galvanized steel, adhesive bonding, interfacial interaction, corrosion resistance, molecular dynamics (MD), density functional theory (DFT)

## Abstract

The epoxy adhesive-galvanized steel adhesive structure has been widely used in various industrial fields, but achieving high bonding strength and corrosion resistance is a challenge. This study examined the impact of surface oxides on the interfacial bonding performance of two types of galvanized steel with Zn–Al or Zn–Al–Mg coatings. Scanning electron microscopy and X-ray photoelectron spectroscopy analysis showed that the Zn–Al coating was covered by ZnO and Al_2_O_3_, while MgO was additionally found on the Zn–Al–Mg coating. Both coatings exhibited excellent adhesion in dry environments, but after 21 days of water soaking, the Zn–Al–Mg joint demonstrated better corrosion resistance than the Zn–Al joint. Numerical simulations revealed that metallic oxides of ZnO, Al_2_O_3_, and MgO had different adsorption preferences for the main components of the adhesive. The adhesion stress at the coating–adhesive interface was mainly due to hydrogen bonds and ionic interactions, and the theoretical adhesion stress of MgO adhesive system was higher than that of ZnO and Al_2_O_3_. The corrosion resistance of the Zn–Al–Mg adhesive interface was mainly due to the stronger corrosion resistance of the coating itself, and the lower water-related hydrogen bond content at the MgO adhesive interface. Understanding these bonding mechanisms can lead to the development of improved adhesive-galvanized steel structures with enhanced corrosion resistance.

## 1. Introduction

Galvanized steel is utilized in various intricate shapes and structures, such as automobile bodies, due to its outstanding corrosion resistance [1,2,3]. In industrial manufacturing, adhesive bonding of metal materials is increasingly replacing conventional joining methods, including welding, diffusion bonding, and riveting, owing to its merits of balanced load, high joint stiffness, and reduced galvanic corrosion between jointed components [4,5,6]. However, the adhesive interface may fail in a humid environment due to moisture-induced desorption [7]. To design high-performance adhesive bonding structures for galvanized steel, it is crucial to understand interfacial interaction and corrosion resistance mechanisms at the atomic and molecular levels. This understanding can also provide theoretical guidance for coating preparation to achieve superior interfacial bonding strength and joint durability.

In recent years, molecular dynamics (MD) simulation and density functional theory (DFT) research has become essential to understand the complex interfacial phenomena between metal and polymer [8,9,10,11]. Bahlakeh et al. [12] investigated the effects of a new cerium–lanthanum (Ce–La) nanofilm-treated steel surfaces on the interfacial bonding mechanism of an epoxy adhesive. They found that the electrostatic interactions between epoxy adhesive and the nanofilm (consisting of CeO_2_ and LaO_3_) were stronger than those on an untreated steel surface. In addition to investigations on flat metal surfaces, some research focuses on the interfacial interactions of metal–adhesive interfaces in nanostructures. Liu et al. [13] used MD simulations to calculate interfacial interactions and bonding processes between polymers and metals (aluminum and copper). They proposed that the viscoelasticity and polarity of the polymers influence the interfacial interactions, which determines the final performance of the bonded structure by influencing the wall-slip behavior. Li et al. [14] investigated the influence of nanopit structures on the interactive behavior and bonding performance between metal and polymer. Compared with rectangular, cylindrical, and pyramidal nanopits, the conical nanopits were found to be beneficial for the bonding performance of the Cu-PPS interface due to their enhanced interfacial energy and wettability. According to the DFT calculation results of Lee et al. [15], the horizontal orientation of epoxy adhesive on Fe (100) metal surface is stronger than the vertical orientation, and the hydroxyl group and benzene ring of the epoxy adhesive are main functional groups that generate adhesion forces in metal–adhesive interface. Moreover, adhesive components adsorbed on the oxide surface significantly affect the minimum energy path and reaction energy. Knaup et al. [16] found that the adhesion promoter, 3-aminopropylmethoxysilane, exhibits a preference for adsorption over bisphenol A diglycidyl ether on the Al_2_O_3_ surface, whereas the adsorption of the curing agent (diethyltriamine) is poor. To better understand the dynamic behavior of polymers on metal surfaces, Semoto et al. [17] showed that hydrogen bonds were generated between the hydroxyl groups of the epoxy polymer and aluminum oxide, which are the main forces contributing to the adhesion force. Additionally, Tsurumi [18] found that the epoxy cresol novolac and phenol novolac fragments form physical bonds to the Cu surface through dispersion forces, while chemical bonds to the surface of Cu_2_O through σ-bonds and hydrogen bonds. The maximum adhesion stress was 1.6 and 2.2 GPa for the Cu and Cu_2_O surfaces, respectively. The hot-dip Zn–Al or Zn–Al–Mg coatings on steel sheets can provide excellent corrosion resistance and alter the adhesion of the steel sheet to adhesive [1,19]. However, existing research is insufficient to explain the interfacial phenomena between the epoxy adhesive and the galvanizing coating, and there is a lack of understanding of the adhesion mechanism of the adhesive interface on the galvanized steel sheet at the molecular level. 

Water at metal–adhesive interface is a common cause of adhesion loss in moist environments, but its effects on metal–adhesive adhesion are not fully understood [7,20,21,22,23]. Semoto et al. [24] proposed that the water molecules in the interface generate a hydrogen bond network and interact with the epoxy resin and the substrate surface, providing a weak adhesion interaction. However, the study of Higuchi et al. [24] on the interface between silica and epoxy resin revealed that adsorbed water molecules reduced the interfacial adhesion energy and force. This reduction may be due to the deformation and flexibility of the H_2_O molecules and the hydrogen bond network. In addition, some scholars have studied the influence of the thickness of water layer on interfacial adhesion energy and force [25]. DFT calculations of the effect of water molecules on the interface between aluminum oxide and epoxy resin revealed that when an H_2_O molecule resides in close proximity to the Al–O bond, it enhances the dissociation of the O atom from the epoxy group, causing the water layer in the interfacial area to become alkaline. This alkaline environment damages the interfacial bonding and breaks the bisA ether groups, leading to a significant reduction in adhesion strength [26,27]. Galvanized steel sheets have complex elements, making it a significant research topic to use the density functional theory (DFT) method to explore the effect of surface oxides on the adhesion loss of the coating–adhesive interface caused by H_2_O molecules.

In this study, it is intended to assess the effect of surface oxides on the interfacial interaction and corrosion resistance between epoxy adhesive and galvanized steel. The physical and chemical properties of Zn–Al and Zn–Al–Mg galvanized steel surfaces were characterized, and the adhesion strength in dry and wet condition between the coatings and the epoxy adhesive was investigated. Molecular models of the adhesive interface were established based on the surface characterization. Through MD simulations, nonbonded interactions between epoxy adhesive molecules and oxide surfaces (consisting of ZnO, Al_2_O_3_, and MgO) were elucidated. Periodic DFT calculations were then carried out on slab models consisting of epoxy fragments and the three oxide surfaces, with and without water molecules at the interface. The calculation results provided electronic characteristics, chemical bonding, and adhesion force information for the interfaces between the adhesives and the oxides. Finally, by combining DFT calculation with EIS test, the corrosion resistance of ZnO, Al_2_O_3_, and MgO adhesive interfaces was compared and analyzed under water conditions.

## 2. Experimental Procedures

### 2.1. Materials

Two types of zinc galvanized steel sheets, with Zn–Al and Zn–Al–Mg coatings and a thickness of 0.8 mm, were selected as the bonding materials in this study. The zinc galvanized steel sheets were cut into 100 × 25 × 0.8 mm^3^ substrates by laser cutting. All specimens were cleaned with anhydrous ethanol before adhesive bonding. Henkel TEROSON EP 5089 (Düsseldorf, Germany), a hot-cured epoxy adhesive (mixing bisphenol A diglycidyl ether, DGEBA, and modified polyurethane resin, MPUR) was selected to prepare the joints. 

### 2.2. Characterization

The chemical compositions of the zinc galvanized steel surfaces were analyzed by using an X-ray photo electron spectroscopy (XPS, ESCALAB 250, Thermo Fisher Scientific (Waltham, MA, USA)). An Al-Kα source operating at 15 kV and 30 mA was used to obtain the XPS spectrum. The basic pressure of the analysis chamber was 1 × 10^−9^ Torr.

A scanning electron microscope (SEM, Nova NanoSEM 450, FEI (Hillsboro, OR, USA)) was used to observe the surface physical morphology of the zinc galvanized steel sheets. Element concentration detection was conducted using an energy dispersive X-ray spectroscope (EDS). 

The open circuit potential and electrochemical impedance spectroscopy (EIS) tests were performed on the Zn–Al and Zn–Al–Mg coating samples using the CorrTest CS310 electrochemical measuring system. A three-electrode system was used, with the galvanized steel sample serving as the working electrode, a platinum sheet electrode (20 mm × 20 mm) as the counter electrode, and Ag/AgCl (saturated KCl) as the reference electrode. The frequency range for the EIS was 10^5^–10^−2^ Hz, during which the open circuit potential (OCP) remained stable. The EIS data were further fitted using the ZView 3.0a software.

### 2.3. Water Soak

To simulate the effect of interfacial water on the interfacial bonding performance, the fully cured galvanized steel/adhesive joints were soaked in deionized water at 55 °C for 21 days, following the standard GMW15200. After the 21-day immersion period, the joints were immediately removed from the water and subjected to lap-shear strength testing. 

### 2.4. Lap-Shear Strength Testing

Lap-shear strength testing was performed to determine the interfacial bonding performance of the joints. The joints were prepared in accordance with ISO4587 with a lapped area of 12.5 × 25 mm^2^. Glass balls with a diameter of 0.25 mm were used to control the thickness of the adhesive layer. Due to the low thickness of the galvanized steel sheet, to avoid its plastic deformation during the joint strength testing, a 1.0 mm thick TC4 titanium alloy sheet was bonded to the back of the lapped side to strengthen the joint. To prevent adverse effects on the adhesive interface caused by corrosion mediums of the galvanized coating in non-lapped area during the water soak, the non-lapped area on the lapped side was painted to protect the galvanized coating. The joints were cured at 170 °C for 20 min to ensure full curing of the adhesive. The schematic diagram and photograph of the joints are shown in Figure 1. Five repeat joints were pulled off on a universal testing machine (MTS E45.105) at a speed of 10 mm/min.

## 3. Modeling Details

The simulation of this work was completed with Materials Studio 2018 software. The Henkel adhesive is a complex epoxy system consisting of epoxy monomers/oligomers, curing agents, and additives. According to the references [9,28,29], the molecular structure of the Henkel adhesive was simplified into two main components: the epoxy monomer/oligomer (DGEBA) and the curing agent (MPUR). The optimized geometries of the monomers of DGEBA, MPUR, and the reaction product of MPUR and DGEBA (abbreviated as MUPR-modified DGEBA) are shown in Figure 2a–c. Surface models of ZnO (10-10), Al_2_O_3_ (100), and MgO (100) were constructed with similar number of atoms of 1296, 1200, and 1296, respectively. After adding a vacuum layer with a thickness of 35 Å on top of the oxide surfaces, the dimensions of the three models were 29.2 Å × 31.2 Å × 50.0 Å, 27.9 Å × 25.2 Å × 49.5 Å, and 26.8 Å × 26.8 Å × 49.7 Å for ZnO (10-10), Al_2_O_3_ (100), and MgO (100), respectively. All adhesive molecular chains were placed 10 Å above the surfaces to ensure a similar initial distance between the adhesive molecules and the oxide surfaces. The main chains were oriented parallel to the surface to maintain the same distance. All oxide–adhesive geometries were optimized and equilibrated for 500 ps with fixed oxide surface atoms at room temperature to ensure that the oxide–adhesive system reached equilibrium. The simulations were carried out under the NVT ensemble and the COMPASS force field [30]. The Andersen thermostat was used for temperature control. By using atom-based cutoff and Ewald methods, nonbonded van der Waals (vdW) and electrostatic interactions were considered, respectively. 

A DGEBA molecule segment was used as the adhesive model in DFT calculations, as shown in Figure 2d. Supercells of ZnO (10-10), Al_2_O_3_ (100), and MgO (100) with 128, 120, and 120 atoms, respectively, and a vacuum layer thickness of 15 Å were placed above the oxide surfaces. The final dimensions of the ZnO (10-10), Al_2_O_3_ (100), and MgO (100) supercells were 13.0 Å × 10.4 Å × 24.4 Å, 12.6 Å × 8.4 Å × 23.4 Å, and 11.2 Å × 8.4 Å × 26.3 Å, respectively. Two adsorption models were established to investigate the effect of water on the interfacial interaction between the adhesive and the galvanized coating: a dry model, in which the DGEBA segment was placed parallel to the oxide surfaces and then geometrically optimized; and a water model, in which the stable structure of five water molecules adsorbed on the surface was calculated, and then the DGEBA segment was placed above the structure of water molecules for geometric optimization. In all adsorption models, the oxide surfaces were fixed, while the adsorbates (adhesive/water molecules) were allowed to relax.

The interaction energy, *E*_int_, in MD simulation and adsorption energy, *E*_ad_, in DFT calculation can be calculated as follows [17]:(1)$$E_{\mathrm{int}\;\mathrm{or}\;\mathrm{ad}}=E_{\mathrm{substrate}/\mathrm{adsorbate}}-(E_{\mathrm{substrate}}+E_{\mathrm{adsorbate}})$$
where *E*_substrate/adsorbate_ represents the total energy of the substrate–adsorbate system, and *E*_substrate_ and *E*_adsorbate_ represent the energies of the substrate and the adsorbate molecule, respectively. 

The adhesion force can be calculated as follows:(2)$$F=\frac{dE_{\mathrm{ad}}}{d\Delta r}$$
where Δ*r* is the distance from the stable equilibrium position of the adhesive molecule, and *E*_ad_ is the total energy of the surface–adhesive system. The value of Δ*r* changes in the range of −0.8~1.2 Å. The system energy containing adhesive molecule and surface was calculated every 0.1 Å in dry/water models. Morse potential approximation fitting of the energy–distance plots was performed using the least square method in the range of −0.8~1.2 Å, where the Morse potential is written as follows:(3)$$E=E_{0}\left(e^{-2\frac{\Delta r}{\lambda }}-2e^{\frac{-\Delta r}{\lambda }}\right)$$
where *E*_0_ is the minimum value of the potential, and *λ* is a constant that determines the range of the interaction force.

All DFT calculations were performed in Dmol^3^ module. The double-numerical plus polarization (DNP) functions were used [31]. A 3 × 3 × 1 k-point grid was used in the calculations. The Perdew–Burke–Ernzerhof (PBE) generalized gradient approximation (GGA) was employed to treat exchange-correlation interactions of electrons and optimize each adsorption mode. The energy and displacement convergence criteria of geometric optimization were set to 2 × 10^−5^ and 5 × 10^−3^, respectively.

## 4. Results and Discussion

### 4.1. Physical and Chemical Properties of Galvanized Steel Surface

The chemical compositions of the Zn–Al and Zn–Al–Mg coatings were investigated using XPS. Table 1 shows that the main metallic elements of the Zn–Al coating were Zn and Al, while the Zn–Al–Mg coating contained an additional 2.16 at. % Mg. The low Fe content on the surface of both coatings indicates complete coverage of the coatings on the steel substrate. The XPS results show that the coating surface contains significant amounts of C and O. C is possibly produced by carbon-containing contaminants during the production, storage, and transportation of galvanized steel. Part of the O element comes from the contaminants and part from metal oxides on the coating. High-resolution XPS spectra of Zn 2p, Al 2p, and Mg 2p measured for Zn–Al and Zn–Al–Mg coatings are presented in Figure 3. The Zn 2p^3/2^ core level of both coatings were deconvoluted into two peaks at 1022.1 eV (oxidized zinc species, Zn^2+^) and 1020.9 eV (metallic zinc, Zn^0^), respectively. The Al 2p peak reflects the oxidation states (Al^3+^) of the Al on the Zn–Al coating, while metallic aluminum (Al^0^) is found in the Zn–Al–Mg coating. The Mg1s has one peak at a binding energy of 1304.1 eV, corresponding to the presence of oxidized magnesium species (Mg^2+^). The composition of these elements is similar to that reported in the literature [1]. Based on the analysis of metal species on the Zn–Al and Zn–Al–Mg coatings, the coatings have a high concentration of metal oxides [32]. Therefore, the surfaces characterized by metal oxides, namely ZnO, Al_2_O_3_, and MgO, will be the focus of the research on the adhesion characteristics between the coatings and epoxy adhesives.

Figure 4 presents SEM and EDS maps of the Zn–Al and Zn–Al–Mg coatings. The Zn–Al coating is mainly composed of a massive microstructure (Figure 4a). As shown in Figure 4b–d, Zn and Al elements are uniformly distributed on the surface, while the higher concentration of O at the edge of the massive microstructure reveals the local enrichment of zinc oxide and aluminum oxide. The Zn–Al–Mg coating presents a mixed surface morphology of convex microstructure and dendritic microstructure (Figure 4e). EDS results show that Zn is uniformly distributed, while Al, Mg, and O are segregated in the dendritic microstructure area [33], as shown in Figure 4f–i, which reflects the uneven distribution of aluminum oxide and magnesium oxide on the surface of the coating.

### 4.2. Interfacial Bonding Performance between Epoxy Adhesive and Galvanized Steel

To evaluate the interfacial bonding performance between Zn–Al or Zn–Al–Mg galvanized steel and adhesive and the effect of water soak on the bonding performance, the lap-shear strengths were measured under both dry and water-soaked conditions. As shown in Figure 5, under dry conditions, the Zn–Al joint exhibits a higher adhesion strength (32.3 MPa) than the Zn–Al–Mg joint (29.2 MPa). Despite the difference in adhesion strengths, both joints show excellent interfacial bonding performance with cohesive fracture surfaces [34,35]. SEM observation of the fracture surface of the Zn–Al–Mg joint reveals a large number of holes, which are few in the fracture surface of the Zn–Al joint. The cross section of the adhesive joint (Figure 5b) reveals the presence of bubbles in the adhesive layer of Zn–Al–Mg joint. In contrast, fewer and smaller bubbles are observed in the adhesive layer of the Zn–Al joint. These bubbles weakened the mechanical properties of the adhesive and contributed to the lower bonding strength of the Zn–Al–Mg joint. The formation of bubbles suggests that the Zn–Al–Mg coating exhibited poor infiltration to the adhesive compared to the Zn–Al coating, and air could not be expelled outside the adhesive structure when the adhesive was in contact with the Zn–Al–Mg coating.

The experimental results indicate that after 21 days of water soaking, the adhesion strength of galvanized steel sheets with Zn–Al and Zn–Al–Mg coatings decreased by 14.7% and 7.6%, respectively. The fracture surface of the Zn–Al joint shows large-area interfacial failure, which is related to the diffusion of corrosive medium from the edge of the interface [19,36,37]. In contrast, less and sporadic interfacial failure areas are observed on the fracture surface of the Zn–Al–Mg joint. These findings suggest that the Zn–Al–Mg coating has better corrosion resistance in a water environment than the Zn–Al coating. To understand the difference in their corrosion resistance, it is crucial to investigate the interfacial interaction between the two coatings and the adhesive, as well as the behavior of water molecules in the interfaces [38].

### 4.3. Molecular Behavior and Adhesion Force at Epoxy Adhesive/Galvanizing Coating Interface

This section focuses on the role of nonbonded and chemical interactions in the interaction between epoxy adhesive and galvanized steel. Nonbonded interactions mainly refer to van der Waals forces and electrostatic interactions, while chemical interactions include ionic bonding, covalent bonding, and coordination bonding, etc. [39]. The physical adsorption resulting from nonbonded interactions can provide some initial adhesion strength, but it is usually not sufficient to form a durable bonding [40]. In contrast, chemical interactions can form a relatively strong and durable bonding between the adhesive and metal surface during the curing process [41].

Upon contact with a metal surface, the adhesive can form a uniform thin film on the surface, driven by nonbonded interactions [41,42]. These forces reduce the distance between the adhesive and the metal surface, promoting infiltration and interfacial contact. Investigating the interfacial nonbonded interactions between metal oxides on the coating surface and epoxy adhesive molecules can help to understand the infiltration behavior of epoxy adhesive on the coating surface at the molecular/atomic scale [43,44,45]. Figure 6 shows the molecular structure and adsorption energy of DGEBA, MPUR, and MPUR-modified DGEBA before and after adsorption on ZnO (10-10) surfaces. After adsorption, all three adsorbents moved to the ZnO substrate, with the distance between the adsorbents and substrate remaining constant as the equilibrium time increased. This process reflects the phenomenon of adhesive infiltration on the ZnO (10-10) surface under the action of nonbonded interaction forces. According to the adsorption results, the ZnO (10-10) surface has adsorption effects on DGEBA, MPUR, and MPUR-modified DGEBA. The simulation of MPUR-modified DGEBA show that the hydroxyl group points to the surface, and the methyl group deviates from the surface, indicating that the hydroxyl group is a functional group that is easily adsorbed, while the methyl group is a functional group that is difficult to be adsorbed. In addition, the adsorption energy of the three adsorbed substances, which is composed of van der Waals and electrostatic interaction energies, is quantitatively analyzed. The nonbonded interaction energies of DGEBA, MPUR, and MPUR-modified DGEBA with ZnO (10-10) surfaces are −49.1 kcal/mol, −42.8 kcal/mol, and −112.6 kcal/mol, respectively. It can be concluded from the comparison of interaction energy that ZnO (10-10) prefers to adsorb DGEBA rather than MUPR. Negative interaction energies further indicate the spontaneous occurrence of the adsorption and infiltration of the adhesive on ZnO (10-10) surfaces, and the van der Waals force dominates the process as demonstrated by the relatively high interaction.

Figure 7a–f shows the molecular structures and adsorption energies of the adsorbates before and after adsorption on the Al_2_O_3_ (100) surface. After adsorption, the adsorbates are found to be close to the Al_2_O_3_ (100) surface, and the distance of the adsorbates in the longitudinal direction decreased, indicating their spreading on the substrate. The nonbonded interaction energies of DGEBA, MPUR, and MPUR-modified DGEBA with the Al_2_O_3_ (100) surface are −67.7 kcal/mol, −61.7 kcal/mol, and −147.1 kcal/mol, respectively. Similar to the ZnO (10-10) surface, the Al_2_O_3_ (100) surface also prefers to adsorb DGEBA. The negative nonbonded interaction energies again demonstrate the interfacial infiltration of the adsorbates on the Al_2_O_3_ (100) surface. Compared with the ZnO (10-10) surface, the Al_2_O_3_ (100) surface has stronger interfacial interactions with the three adsorbates, which is confirmed by the higher interaction energy. However, considering that chemical interactions also play a crucial role in determining the final adhesion effect, this result does not necessarily indicate that the adhesive force on the Al_2_O_3_ (100) surface is stronger than that on the ZnO (10-10) surface.

The molecular structures and adsorption energies of DGEBA, MPUR, and MPUR-modified DGEBA on the MgO (100) surface before and after adsorption are shown in Figure 8. After adsorption, all three adsorbents are found to be in close proximity to the MgO substrate. However, unlike on the ZnO (10-10) and Al_2_O_3_ (100) surfaces, the longitudinal size of MPUR-modified DGEBA did not decrease significantly after adsorption on the MgO (100) surface, indicating that its infiltration into the MgO surface is lower. The nonbonded interaction energies of DGEBA, MPUR, and MPUR-modified DGEBA with the ZnO (10-10) surface are −40.7 kcal/mol, −61.6 kcal/mol, and −89.2 kcal/mol, respectively. Compared with DGEBA, the MgO (100) surface shows a stronger preference for the adsorption of MPUR due to its electrostatic force. Furthermore, the difference in interaction energy between DGEBA and MPUR on the MgO (100) surface (51.4%) is significantly higher than that on the ZnO (10-10) (12.8%) and Al_2_O_3_ (100) (8.9%) surfaces, indicating a more obvious preference for MPUR adsorption on the MgO (100) surface. However, as previously analyzed, the adhesion force of the adhesive on the coating surface is not only directly determined by the nonbonded interaction but also by chemical interaction.

To quantitatively investigate the influence of ZnO, Al_2_O_3_, and MgO on the adhesion performance of the galvanized steel sheet, DFT calculations were employed to evaluate the adhesion force and chemical bonding of DGEBA fragment on three oxide surfaces. In a dry environment, the calculated energy–displacement curve is shown in Figure 9a. The curve conforms to the Morse potential in the range of −0.8 eV to 1.2 eV. The distance between the DGEBA fragment and the surface was defined as 0 Å, where the system reached a stable adsorption state. As the distance exceeds 0, the adsorption energy tends to 0, and as the distance becomes less than 0, the adsorption energy tends to infinity. The force–distance curves are shown in Figure 9b. To compare *F*_max_ with the macroscopic adhesion strength, *F*_max_ was converted into adhesion stress, which can be calculated as follows:(4)$$S_{\max}=F_{\max}/A$$
where *A* is mean value of surface areas of oxide models, 1.12 × 10^−18^ m^−2^.

The fitting parameters obtained are shown in Table 2. As can be seen, the adhesion stresses of the DGEBA fragment on ZnO(10-10), Al_2_O_3_(100), and MgO(100) surfaces are 0.96 GPa, 0.68 GPa, and 1.14 GPa, respectively, which are in good agreement with the order of magnitude reported in the literature [17,18,25]. The calculated adhesion stresses are two orders of magnitude larger than the actual joint strengths (Zn–Al: 32.3 MPa, Zn–Al–Mg: 29.2 MPa). This difference is attributed to the fact that the actual joint strength is affected by more complex factors, such as surface micro-nano structure, surface contaminants, internal stress, adhesive infiltration on the bonded surface, etc. [46,47,48]. Moreover, it was found that cohesion failure occurred in both the experimental Zn–Al and Zn–Al–Mg joints, but the true value of the interfacial strength was not obtained. Therefore, the calculated adhesion stress is a theoretical value of the interfacial strength, which is greater than the actual joint strength. As previously analyzed, the calculated adhesion stress between the adhesive and MgO is significantly higher than that of the ZnO and Al_2_O_3_ models, which is a significant factor contributing to the excellent interfacial bonding performance of the Zn–Al–Mg joint.

To elucidate the chemical interaction between the adhesive and the oxide substrates, the molecular structure and difference charge density plots of the ZnO, Al_2_O_3_, and MgO systems after chemisorption equilibrium were analyzed [49], as shown in Figure 10. The adhesion behavior of the adhesive molecules was found to be different on the three oxide surfaces. The straight adhesive molecules are adsorbed on the ZnO (10-10) and MgO (100) surfaces at angles of 7.4° and 4.6°, respectively, while the bent adhesive molecule is adsorbed on the Al_2_O_3_ (100) surface. It is well known that the more parallel the adhesive is to the surface, the more atoms participate in the interfacial chemical interaction, which is the reason why the MgO (100) surface can obtain a relatively higher adhesion force. In addition, the adhesive molecules generated different chemical bonding with the three surfaces. On the ZnO (10-10) surface, the O atom on the surface is surrounded by the electron accumulation region, while the H atom of hydroxyl group in the adhesive is surrounded by the electron depletion region, which indicates the formation of hydrogen bond. On the Al_2_O_3_ (100) surface, the 5-coordinated Al atom loses electrons and is surrounded by electron depletion regions, while the O atom of ether group is surrounded by electron accumulation regions. Based on these observations, it can be concluded that there is ionic interaction between the 5-coordinated Al atom and the O atom of the adhesive. In addition, a stronger ionic interaction was generated between the MgO (100) surface and the hydroxyl group of the adhesive than in the ZnO model, which is demonstrated by the darker color of the electron accumulation and depletion regions between the Mg atoms and hydroxyl O. The above analysis shows that hydrogen and ionic bonds formed between the adhesive and oxides are the origin of interfacial adhesion force. Compared with the ZnO (10-10) and Al_2_O_3_ (100) surfaces, the MgO (100) surface has a chemical adsorption advantage on adhesive molecules in terms of molecular configuration and electronic interaction, which is the reason for its high adhesion.

### 4.4. Corrosion Resistance Mechanism at Epoxy Adhesive/Galvanizing Coating Interface

The immersion and diffusion of water molecules at the adhesive interface, as well as their corrosion on the coating surface, can accelerate the degradation of interfacial bonding performance [50]. To assess the corrosion resistance of Zn–Al and Zn–Al–Mg coatings, the electrochemical impedance spectroscopy (EIS) data were analyzed. Figure 11a shows the Nyquist plots of the two coatings, which take the form of a semicircle, indicating the occurrence of corrosion on the coating surfaces. The radius of the Nyquist curve is indicative of the corrosion resistance of the coating, with a larger radius corresponding to a lower corrosion rate and stronger corrosion resistance. The Nyquist curve radius of the Zn–Al–Mg coating is greater than that of the Zn–Al coating, demonstrating that it possesses stronger corrosion resistance. The equivalent circuit diagram of the EIS fitting is shown in Figure 11b, where R_s_ represents the resistance of the corrosive solution, R_ct_ represents the resistance of charge transfer, and a constant phase element (CPE) replaced the capacitive element of coating to obtain the best fit [51]. The fitting data for the EIS equivalent circuit are provided in Table 3, where the R_s_ values for both samples are similar (with a relative error of no more than 5%). CPE-T and CPE-P represent the characteristic and index parameters of the CPE, respectively [52]. In comparison to the Zn–Al coating, the Zn–Al–Mg coating exhibits a lower CPE-T value and a higher R_ct_ value, which further indicates its stronger corrosion resistance. As a result, it is relatively difficult for water molecules to cause corrosion at the adhesive interface of the Zn–Al–Mg coating compared to the Zn–Al coating, resulting in better interfacial bonding performance between the Zn–Al–Mg coating and the adhesive.

To further investigate the corrosion resistance mechanism of the adhesive interface on galvanized steel sheets, the interfacial adhesion forces of oxide–adhesive systems were calculated in a water environment. The energy–displacement and force–displacement curves of the ZnO, Al_2_O_3_, and MgO adhesion systems in water are shown in Figure 9c,d, and the fitting parameters, maximum adhesion force, and adhesion stress of the energy–displacement curves are summarized in Table 2. As shown in Figure 9c,d, the force–displacement curves of the three oxide systems are significantly reduced in the presence of water molecules, indicating a decrease in interfacial bonding performance. The adhesion stress of the ZnO, Al_2_O_3_, and MgO systems decreased by 21.9%, 26.5%, and 28.9%, respectively. This reduction reflects the sensitivity of adhesion strength to water molecules, with the ZnO system showing the lowest sensitivity, followed by the Al_2_O_3_ system, and the MgO system showing the highest sensitivity. Even with a decrease in adhesion stress, the MgO system still exhibited the highest adhesion stress among the three systems, which is mainly due to the molecular structure of the interface.

Figure 12 displays the stable adsorption structure of the adhesive on the oxide surfaces in the water model. As illustrated in Figure 12, the presence of water molecules in the interface increased the spatial distance between adhesive molecules and the oxide surfaces, which negatively impacted the generation of adhesion force at the interface. Conversely, hydrogen bonds formed between water molecules and O atoms of adhesives and oxides, creating an interfacial hydrogen-bonding network. This network broke the chemical interaction between the adhesives and the oxide surfaces, thus reducing the adhesion performance of the coatings. The hydrogen bond network contents at the three oxide–adhesive interfaces were quantitatively analyzed as the number of hydrogen bonds divided by the number of hydrogen atoms in water molecules. It was found that the value was lower in the MgO–adhesive interface (0.4) than in the ZnO–adhesive (0.8) and Al_2_O_3_–adhesive (0.7) interfaces, resulting in higher adhesion stress in the MgO system than in the ZnO and Al_2_O_3_ systems. 

Based on the above analysis, the addition of Mg element in the coating has a positive effect on the corrosion resistance of the adhesive interface, which is in agreement with the experimental results that indicate a stronger adhesive interface corrosion resistance of the Zn–Al–Mg coating compared to the Zn–Al coating. As shown in Figure 3, the dendritic structure containing magnesium is extensively distributed on the surface of the Zn–Al–Mg coating, providing numerous sites for the interaction between the coating and the adhesive. Water molecules are relatively difficult to diffuse at MgO–adhesive interface. Moreover, the corrosion resistance of the Zn–Al–Mg coating is better than that of the Zn–Al coating, thus further improving the corrosion resistance of the Zn–Al–Mg adhesive interface when compared to the Zn–Al coating adhesive interface.

## 5. Conclusions

This study investigated the interfacial interaction of epoxy adhesive on galvanized steel (Zn–Al and Zn–Al–Mg) and its interfacial corrosion resistance through a combination of experiments and simulations. It was found that the bubbles in the adhesive layer of the Zn–Al–Mg joint reduced its mechanical properties, resulting in the joint strength being approximately 10% lower than that of the Zn–Al joint in the case of cohesive failure. However, the Zn–Al–Mg joint has better interfacial corrosion resistance than the Zn–Al joint after 21 days water soak. Since the coating surfaces were shown to be covered by metal oxides (ZnO, Al_2_O_3_ and MgO), the influence of these oxides on the interfacial interaction and corrosion resistance between galvanized steel and adhesive was analyzed. The nonbonded interaction analysis revealed that the ZnO and Al_2_O_3_ surfaces preferentially adsorbed epoxy resin, while the surface of MgO showed a preference for adsorbing curing agent. The hydrogen bond and ionic interaction between oxide surfaces and the adhesive contributes to the interfacial adhesion stress. The theoretical adhesion stress of MgO is the highest and 1.19 times and 1.68 times of ZnO and Al_2_O_3_, respectively. The corrosion resistance of the Zn–Al–Mg coating was stronger than that of the Zn–Al coating, as demonstrated by the better EIS results. MgO in the Zn–Al–Mg coating is conducive to the reduction of the hydrogen bond network content related to water molecule at the coating–adhesive interface, resulting in better interfacial corrosion resistance.

## Figures and Tables

**Figure 1 materials-16-03061-f001:**
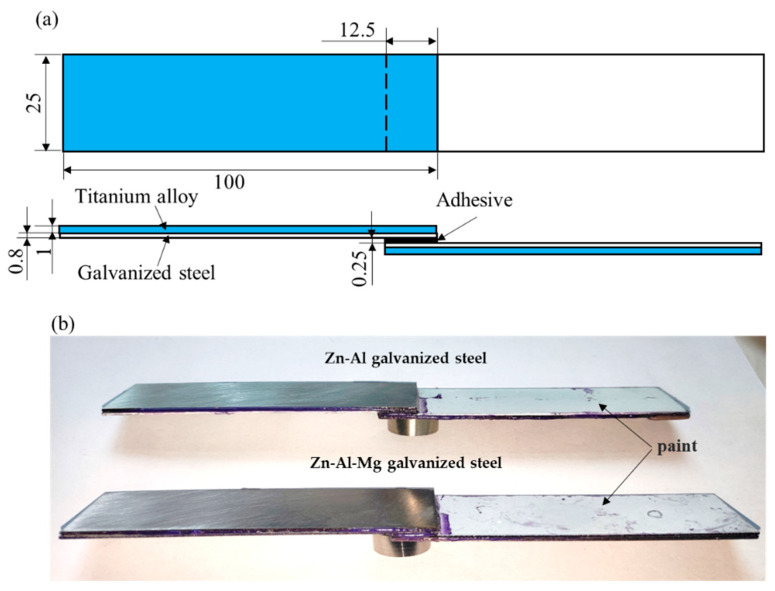
(**a**) Schematic diagram and (**b**) photograph of the galvanized steel joints.

**Figure 2 materials-16-03061-f002:**
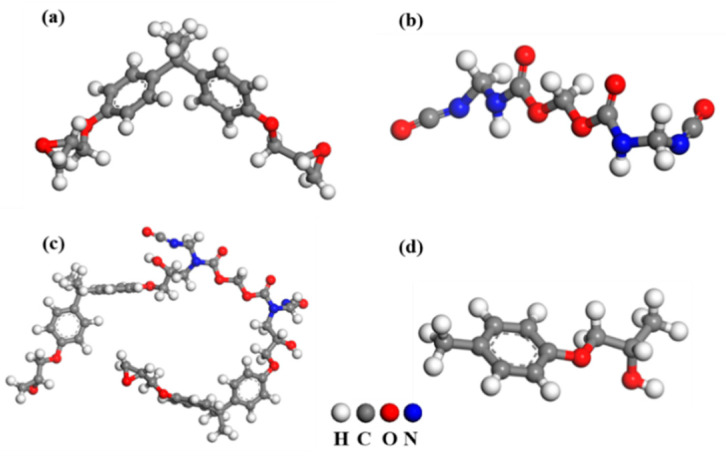
Optimized geometries of the monomer of (**a**) DGEBA, (**b**) MUPR, and (**c**) MUPR-modified DGEBA in MD simulation, and (**d**) DGEBA segment in DFT calculation [16,28,29].

**Figure 3 materials-16-03061-f003:**
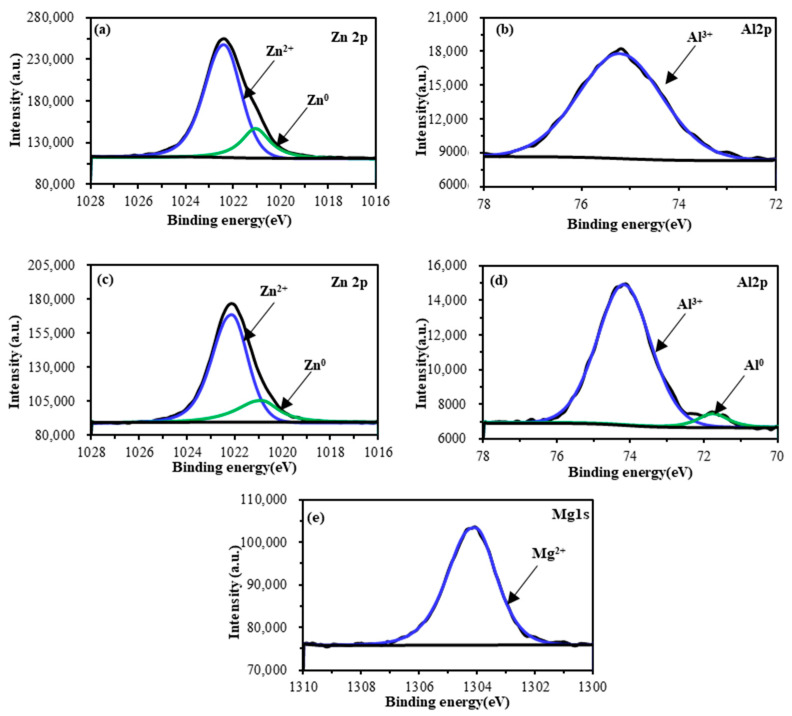
High resolution elemental spectra of (**a**) Zn 2p, (**b**) Al 2p of the Zn–Al coating, (**c**) Zn 2p, (**d**) Al 2p, and (**e**) Mg 1s of the Zn–Al–Mg coating.

**Figure 4 materials-16-03061-f004:**
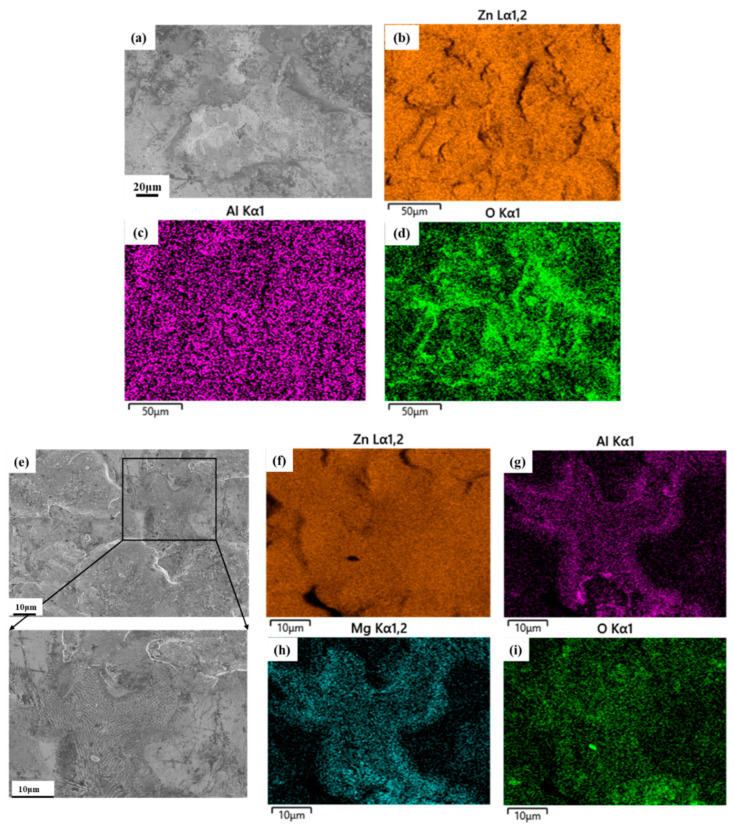
SEM micrographs and EDS maps of elements acquired on (**a**–**d**) Zn–Al and (**e**–**i**) Zn–Al–Mg coating surfaces.

**Figure 5 materials-16-03061-f005:**
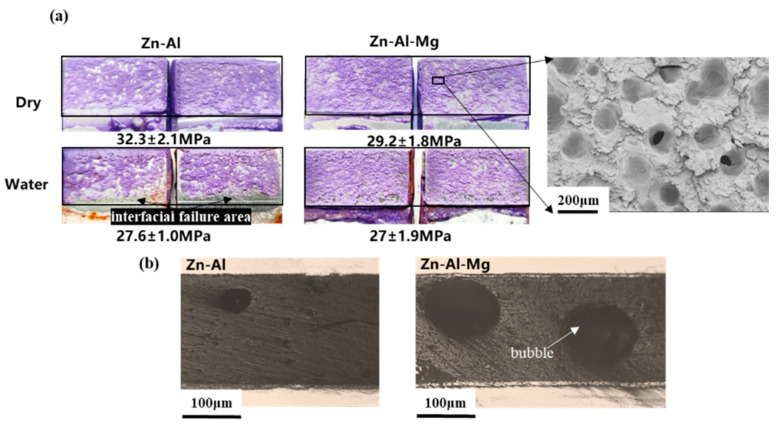
(**a**) Adhesion strength and fracture surfaces of dry and water joint and (**b**) cross section of dry joints of the Zn–Al and Zn–Al–Mg galvanized steel sheets.

**Figure 6 materials-16-03061-f006:**
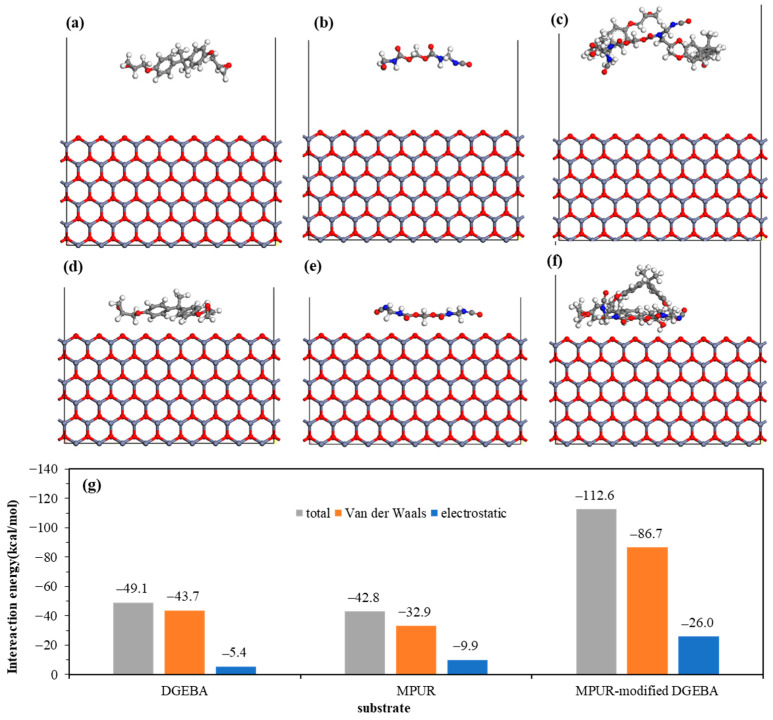
Molecular structure of ZnO(10-10) surface before and after interaction with (**a**,**b**) DGEBA, (**c**,**d**) MPUR, and (**e**,**f**) MPUR-modified DGEBA, and (**g**) their adsorption energy.

**Figure 7 materials-16-03061-f007:**
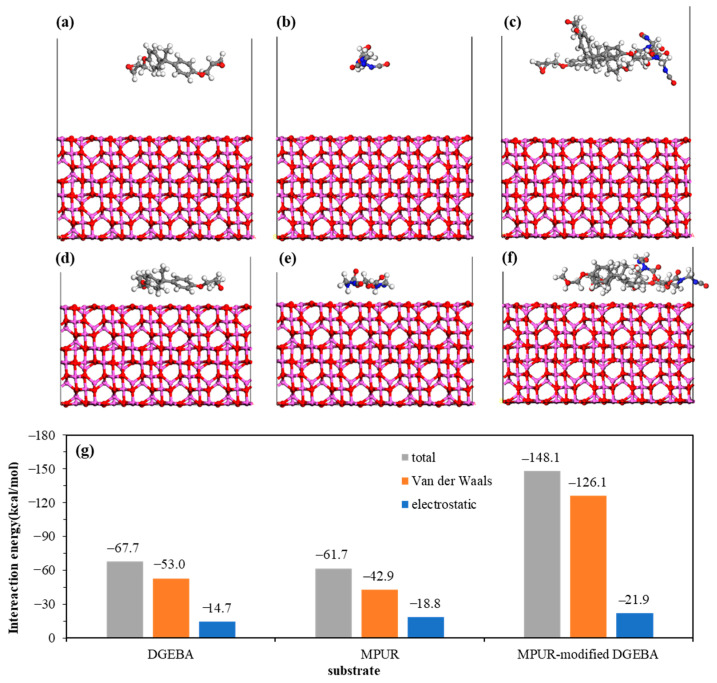
Molecular structure of Al_2_O_3_ (100) surface before and after interaction with (**a**,**b**) DGEBA, (**c**,**d**) MPUR, and (**e**,**f**) MPUR-modified DGEBA, and (**g**) their adsorption energy.

**Figure 8 materials-16-03061-f008:**
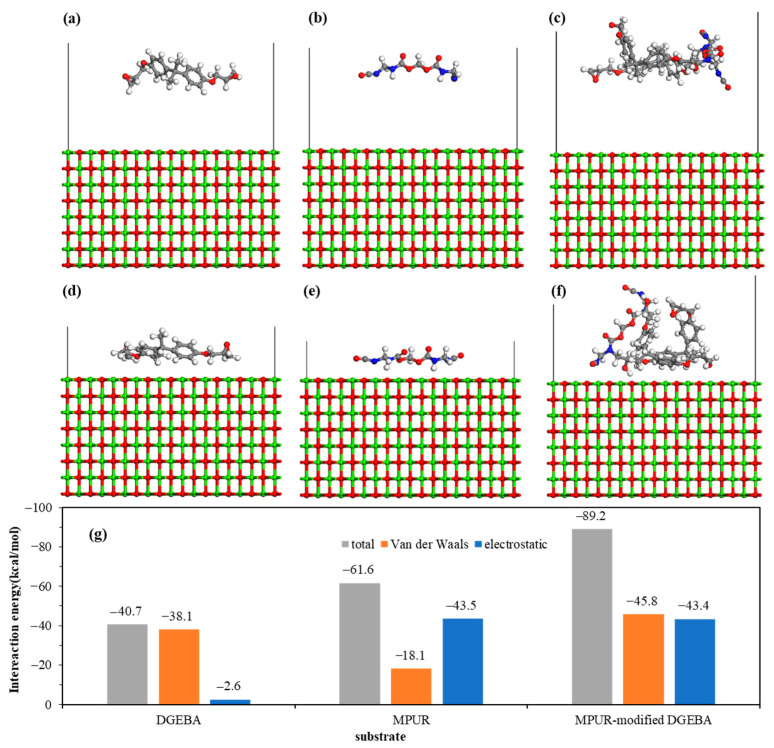
Molecular structure of MgO(100) surface before and after interaction with (**a**,**b**) DGEBA, (**c**,**d**) MPUR, and (**e**,**f**) MPUR-modified DGEBA, and (**g**) their adsorption energy.

**Figure 9 materials-16-03061-f009:**
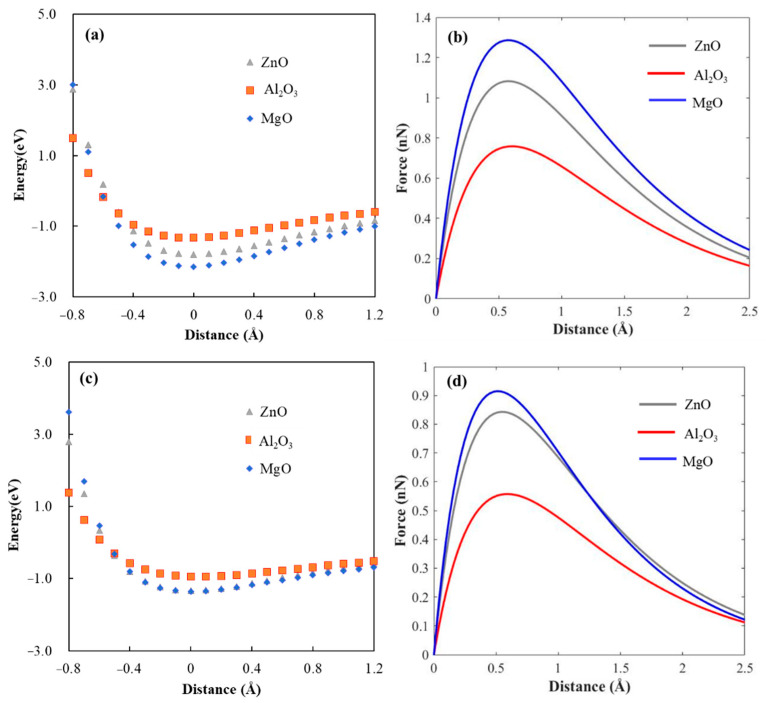
Energy−displacement plots and force−displacement plots for (**a**,**b**) dry environment and (**c**,**d**) water environment.

**Figure 10 materials-16-03061-f010:**
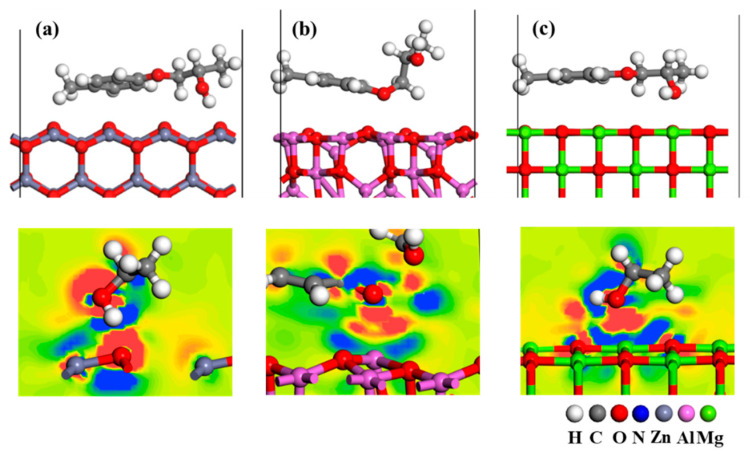
Molecular structures and difference charge density plots calculated for the adhesive interface of (**a**) ZnO, (**b**) Al_2_O_3_, and (**c**) MgO models (electron failure depletion = blue; electron accumulation = red).

**Figure 11 materials-16-03061-f011:**
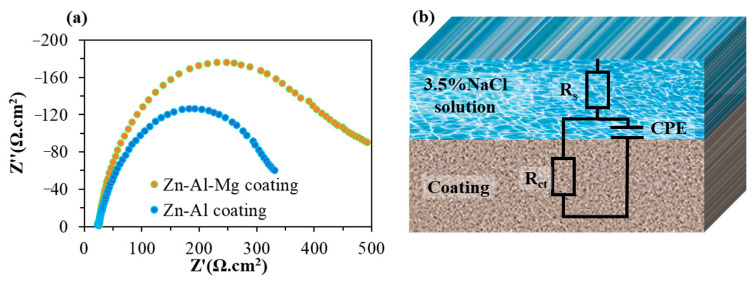
(**a**) Nyquist plot and (**b**) equivalent circuit used to fit the EIS data of the galvanized coating surface.

**Figure 12 materials-16-03061-f012:**
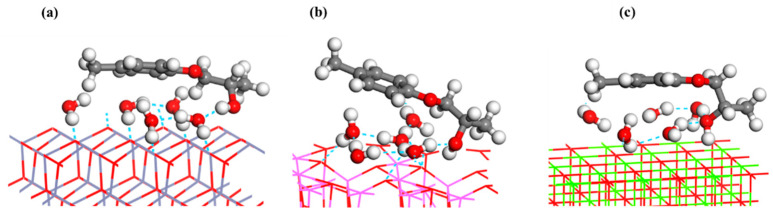
Stable adsorption structure of adhesive on (**a**) ZnO(10-10), (**b**) Al_2_O_3_(100), and (**c**) MgO(100) surfaces in the water model.

**Table 1 materials-16-03061-t001:** Chemical composition (at. %) on the surface of Zn–Al and Zn–Al–Mg coatings.

Element	Atomic %	
	Zn–Al	Zn–Al–Mg
Zn	13.10	8.37
Al	9.00	7.41
Mg	0.05	2.16
Fe	0.69	0.36
O	37.47	42.27
C	39.70	39.43

**Table 2 materials-16-03061-t002:** Theoretical adhesion properties and fitting parameters for the dry and water surface models.

Model	Oxides in Coating	*E*_0_/eV	λ/Å^−1^	*F*_max_/nN	*S*_max_/GPa in This Work	*S*_max_/GPa in Reference
dry	ZnO	1.80	0.8331	1.08	0.96	0.62 [17], 0.61 [17], 1.55 [18], 2.18 [18], 1.546 [25]
	Al_2_O_3_	1.33	0.8748	0.76	0.68	
	MgO	2.14	0.8324	1.28	1.14	
water	ZnO	1.34	0.7929	0.84	0.75	0.51 [24], 0.50 [24], 1.38 [46]
	Al_2_O_3_	0.95	0.8512	0.56	0.50	
	MgO	1.36	0.7413	0.91	0.81	

**Table 3 materials-16-03061-t003:** EIS equivalent fitting data.

Sample	R_s_ (Ω·cm^2^)	R_ct_ (Ω·cm^2^)	CPE-T (s^-n^/Ω·cm^2^)	CPE-P	χ^2^
Zn–Al coating	23.65	342.10	4.66 × 10^−5^	0.79	0.0007
Zn–Al–Mg coating	24.83	461.10	3.24 × 10^−5^	0.85	0.0048

## Data Availability

The data can be provided by authors on request.

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
