# Peer review of "Molecular Understanding of the Interfacial Interaction and Corrosion Resistance between Epoxy Adhesive and Metallic Oxides on Galvanized Steel"

_materials, 2023, doi:10.3390/ma16083061_

Round 1
Reviewer 1 Report
1. Plagiarism of the paper is checked.
2. Avoid using long sentences. Eg, line 34-37, 41-45, 56-60
3. It is suggested to check English language and grammar. Avoid use of first person pronouns. Eg, line 98-100, 102-104
4. For section 3: Modelling details wherever needed mention references. Also, figures and tables if required mention references.
5. Results of experiments obtained are claimed. It is strongly suggested to discuss obtained results/claims with cross reference citation.
Author Response
Thank you for your comments and suggestions. Please refer to the attachment for relevant modifications.

Reviewer 2 Report
The effects of surface oxides on the interfacial bonding properties of two types of galvanized steel were investigated by the authors. SEM and X-ray photoelectron spectroscopy revealed that ZnO, Al2O3, and MgO had different adsorption preferences for the adhesive's main components. The adhesion force at the coating adhesive interface was found to be primarily due to hydrogen bonds and ionic interactions, according to molecular dynamics simulations. Understanding these bonding mechanisms can lead to better adhesive-galvanized steel structures that are more corrosion-resistant. However, the manuscript cannot be published in this form. This research should be improved into a corrosion resistance test using the EIS method to prove that the adhesive material interacts with the metal surface; this part needs more time to be done before publication can be done. The title should be revised to be more concise. The conclusion should be revised into paragraphs and not point by point.
I recommend rejecting this form.
Author Response
Authors’ Response: Thank you for your comments. The special issue of Materials magazine titled "Surface Engineering & Coating Technologies for Corrosion and Tribocorrosion Resistance" has a key focus on surface engineering, coating technologies, and corrosion resistance. Our study investigates the corrosion resistance mechanism on the adhesive interface of galvanized steel sheets, which involves surface coating technology and the corrosion resistance of galvanized steel sheets. Our research aligns perfectly with the scope of this special issue, making it a contribution to the field of surface engineering and coating technologies for corrosion resistance.
Reviewer 3 Report
28-2-23 review report
Journal
Materials (ISSN 1996-1944)
Manuscript ID
materials-2260708
Article 1
Molecular understanding of the interfacial interaction and cor- 2 rosion resistance between epoxy adhesive and metallic oxides 3 on galvanized steel 4
Shuangshuang Li1, Yanliang Zhao2, Hailang Wan1*, Jianping Lin1* and Junying Min1
Oooooooooooooooooooooooooo
- The epoxy adhesive-galvanized steel adhesive structure has been
widely used in various industrial fields, but achieving high bonding strength
and corrosion resistance is a challenge. - The authors have examined the impact of surface oxides on the interfacial bonding properties of two types of galvanized steel with Zn-Al or Zn-Al-Mg coatings.
- Scanning electron microscopy and X-ray photoelectron spectroscopy analysis revealed that the Zn-Al coating was covered by ZnO and Al2O3 , while MgO was additionally found
on the Zn-Al-Mg coating. - Both coatings exhibited excellent bonding in dry environments, but after 21 days of water soaking, the Zn-Al-Mg joint demonstrated better corrosion resistance than the Zn-Al joint.
- Molecular dynamics simulations revealed that metallic oxides of ZnO, Al2O3 and MgO had different adsorption preferences for the main components of the adhesive, and
density functional theory calculations showed that the adhesion force at the
coating-adhesive interface was mainly due to hydrogen bonds and ionic
interactions. - Introduction part is good. Some new references may be added.
- The experimental part is discussed in detail.
- Figures are very good. And colorful
- The experimental data are scientifically interpreted
- And useful conclusions useful to scientific society are derived
- The English of the paper may be improved
- The paper may be accepted for publication after minor revision
- OOOOOOOOOOOOOO
|
1 |
Article Synthesis, characterization and adsorption behavior of modified cellulose nanocrystals towards different cationic dyes
|
Aziz, T., Farid, A., Chinnam, S., ...Habila, M.A., Akhtar, M.S. |
Chemosphere, 321, 137999 |
2023 |
0 |
|||||
|
|
|
|
|
|||||||
|
|
||||||||||
|
|
|
|
|
|
||||||
|
|
||||||||||
|
2 |
Article Enhanced active corrosion protection coatings for aluminum alloys with two corrosion inhibitors co-incorporated in nanocontainers |
Ma, L., Wang, J., Wang, Y., ...Fu, D., Zhang, D. |
Corrosion Science, 208, 110663 |
2022 |
4 |
|||||
|
View at Publisher Related documents |
||||||||||
|
|
||||||||||
|
|
||||||||||
|
|
||||||||||
|
|
||||||||||
|
|
||||||||||
|
|
||||||||||
|
|
||||||||||
|
|
||||||||||
|
3 |
Article • Open access Molecular origins of Epoxy-Amine/Iron oxide interphase formation |
Morsch, S., Wand, C.R., Emad, S., ...Eichhorn, K.-J., Gibbon, S. |
Journal of Colloid and Interface Science, 613, pp. 415–425 |
2022 |
6 |
|||||
|
View at Publisher Related documents |
||||||||||
|
|
||||||||||
Author Response

(The authors gave the same response as above.)

Reviewer 4 Report
- Repeated formatting errors, especially where subscripts are used.
- Language errors, e.g. in line 115 "gluing".
- The paper lacked photographs of the actual samples tested. How adhesive joints were made?
Author Response
Thank you for your comments and suggestions. Please refer to the attachment for relevant modifications
